evolution

ageing, dietary restriction, lifespan extension, senescence

**Authors for correspondence:**
Tracey Chapman
e-mail: tracey.chapman@uea.ac.uk
Alexei A. Maklakov
e-mail: A.Maklakov@uea.ac.uk

# Fitness benefits of dietary restriction

Zahida Sultanova, Edward R. Ivimey-Cook, Tracey Chapman and Alexei A. Maklakov

School of Biological Sciences, University of East Anglia, Norwich Research Park, Norwich NR4 7TU, UK

ZS, 0000-0003-3224-8812; ERI-C, 0000-0003-4910-0443; TC, 0000-0002-2401-8120; AAM, 0000-0002-5809-1203

Dietary restriction (DR) improves survival across a wide range of taxa yet remains poorly understood. The key unresolved question is whether this evolutionarily conserved response to temporary lack of food is adaptive. Recent work suggests that early-life DR reduces survival and reproduction when nutrients subsequently become plentiful, thereby challenging adaptive explanations. A new hypothesis maintains that increased survival under DR results from reduced costs of overfeeding. We tested the adaptive value of DR response in an outbred population of *Drosophila melanogaster* fruit flies. We found that DR females did not suffer from reduced survival upon subsequent re-feeding and had increased reproduction and mating success compared to their continuously fully fed (FF) counterparts. The increase in post-DR reproductive performance was of sufficient magnitude that females experiencing early-life DR had the same total fecundity as continuously FF individuals. Our results suggest that the DR response is adaptive and increases fitness when temporary food shortages cease.

## 1. Introduction

Dietary restriction (DR), reduced food intake without malnutrition, extends lifespan and/or improves health across a broad variety of organisms, from yeast to invertebrates to mammals, including humans [1–4]. However, DR also reduces reproduction, which raises questions about its evolutionary origins [5–7] and presents a potential drawback for the application of DR to maintain human health [3,8].

The DR response—increased survival and reduced reproduction—can be adaptive if organisms choose to reduce investment in reproduction in a nutritionally poor environment and wait for resources to become plentiful [9,10]. The increase in survival as a result of DR could result from (i) allocation of limited resources to somatic maintenance and (ii) reduced costs of reproduction. There is some evidence that dietary restricted organisms allocate more resources to somatic maintenance as shown previously in *Drosophila* [11]. However, it is likely that reduced direct costs of mating and reproduction, e.g. reduced physiological damage, also play an important role [12]. It has been shown previously in *Drosophila* and *C. elegans* that the perception of food availability plays an important role in nutrient-sensing signalling and can mediate a DR response [13–15]. We recently showed that perception of nutrient availability mediated by food odour increases investment in reproduction and decreases survival in *C. elegans* [16]. This suggests that DR worms may 'choose' to delay reproduction in unfavourable conditions, and lay fewer eggs than they potentially could, thus minimizing the potential for starvation that would otherwise occur if offspring emerged into a resource-lacking environment that cannot support development. This finding supports the notion that the DR response is an adaptive life-history strategy.

However, there are potential costs associated with lifespan extension via DR that challenge the adaptive explanation [5]. Recent work in *Drosophila melanogaster* indicated that a return to an *ad libitum* diet after a period of DR was associated with increased mortality and reduced reproduction [17]. While

increased mortality is expected if flies switch from somatic preservation to reproduction, the combined effect of reduced survival and reduced reproduction suggested significant, underappreciated costs of DR. This prompted a reappraisal of the life-history theory-based explanation of the DR response [17]. An alternative hypothesis suggested that lifespan extension under DR avoids the costs associated with a nutrient-rich environment, specifically, detrimental effects of dietary protein on survival [17]. These findings are in line with a recent experimental evolution study suggesting that the link between increased lifespan and reduced reproduction under DR can be uncoupled [7,18].

Understanding whether DR improves or worsens post-DR reproductive performance and fitness is key for developing cogent evolutionary models of the DR response. It also has important implications for translational research aimed at developing nutrient intake regimes and DR mimetics to improve human health. Here we explored the effect of DR on post-DR mortality rates, reproduction and fitness in an outbred population of *D. melanogaster* fruit flies while controlling for possible effects of food odour. In line with previous work, we found that the Gompertz rate parameter ($b1$) increased upon a return to *ad libitum* food conditions after DR. However, we also found that, following a period of DR, females showed increased mating behaviour and reproductive output following their return to *ad libitum* food. Strikingly, this increased reproductive output fully compensated for reduced reproduction during the DR phase.

## 2. Methods

### (a) Experimental population

We used *Drosophila melanogaster* flies from an outbred laboratory-adapted, wild-type Dahomey population. This was derived from an original population founded in 1970 [19]. The population is maintained with overlapping generations at 25°C, approximately 50–60% humidity and a 12 h : 12 h light : dark cycle, and fed with standard sugar yeast agar (SYA) medium (100 g brewer's yeast powder, 50 g sugar, 15 g agar, 30 ml Nipagin (10% w/v solution) and 3 ml propionic acid, per litre of medium). To obtain experimental flies, we collected larvae from eggs derived from females housed in our population cages. Eggs were collected by using grape-agar filled Petri dishes with a smear of live yeast paste. Larvae were raised at a standard density of 100 per vial (glass vials, 25 mm diameter × 75 mm high) each containing 7 ml SYA medium. Virgin adults emerging from these larvae were collected within 7 h of eclosion using ice anaesthesia. The females were collected in same-sex groups of 20 before being set-up in different experimental treatments. Males were collected weekly and maintained in same-sex groups of 20 until their use in mating assays.

### (b) Dietary restriction and control diet treatments

Two days after emergence, females and males were placed together in bottles for 24 h so that all the females could mate (50 females and 50 males were placed in each bottle). Females were then randomly allocated to the different diet treatments and placed individually in the treatment vials (figure 1). All vials contained 40% SYA medium (40 g brewer's yeast powder, 50 g sugar, 15 g agar, 30 ml Nipagin (10% w/v solution) and 3 ml propionic acid, per litre of medium) and a central perforated acetate divider through which females could receive olfactory and visual cues but which prevented their passage to the other side of the vial (figure 1). Four different diets were prepared: fully fed (FF), DR, DR with odour (DR$_{od}$) and FF with odour (FF$_{od}$). The FF diet consisted of 40% SYA

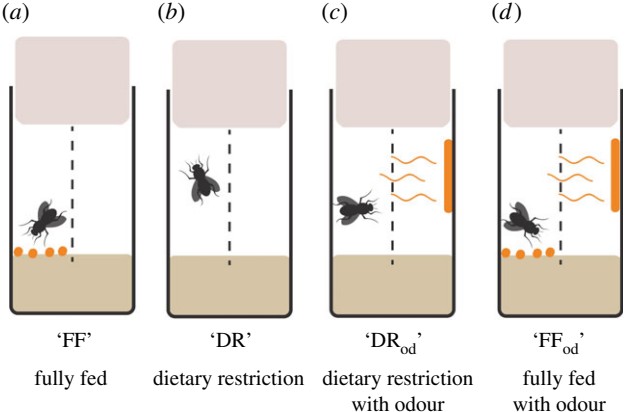

**Figure 1.** Schematic illustration of the different dietary treatments. All vials contained 40% SYA medium and a perforated acetate divider (dashed line) to allow the passage of odours, but not of females to the other side. The FF diet (*a*) comprised the 40% SYA base medium with extra yeast particles added, to which females had *ad libitum* access. The DR diet (*b*) had the same based medium but no added live yeast. DR$_{od}$ and FF$_{od}$ diets (*c,d*) were similar to DR and FF, respectively, but females were exposed to the sight and odour of live yeast by adding a smear of live yeast paste to the other side of the perforated acetate (shown in orange).

medium with excess yeast particles and the DR treatment 40% SYA medium only. We added eight granules of yeast particles per vial, which is more than flies could consume until they were transferred to a new vial. DR$_{od}$ and FF$_{od}$ diets were similar to DR and FF diets, respectively, but females were exposed to the sight and odour of live yeast by adding a smear of live yeast paste to the other side of the perforated acetate (figure 1). The experiment comprised four continuous diet treatments (i.e. constant FF, constant DR, constant DR$_{od}$ and constant FF$_{od}$) and four switch diet treatments (FF-to-DR, DR-to-FF, FF-to-DR$_{od}$, DR$_{od}$-to-FF). Four hundred female flies were monitored individually throughout their lifetime (original starting sample size of $n \approx 50$ per treatment; precise sample sizes per treatment shown in the electronic supplementary material, figure S4). The diet switching took place on day 19 when more than 80% of females were still alive. Throughout the experiment, females were transferred to a new vial three times a week by using $CO_2$ anaesthesia. Deaths and censors were recorded daily.

### (c) Mating and fitness assays of females subjected to the dietary restriction and control, continuous and diet switching treatments

Each focal female was put together with a young wild-type male (1-week old) for 24 h every 7 days. Behavioural observations were conducted in a 25°C room, starting 1 h after lights on (at 10 : 30) for a period of 4 h. Flies that did not mate within the first 4 h were considered non-maters (i.e. mating success = 0). Females and males were kept together for a total of 24 h. At the end of this time, males were discarded. Females were then transferred into new vials, where they laid eggs for another 24 h, and then transferred again. The vacated vials in which females had laid eggs were immediately frozen for subsequent egg counting (i.e. fecundity). This gave a measure of age-related fitness and a proxy for lifetime reproductive success (LRS).

### (d) Statistical analysis

All analyses were performed in R v. 3.3.2 [20]. To test how age-specific survival was affected by dietary treatment, we used the Bayesian survival trajectory analysis implemented using the 'BaSTA' package v 1.9.5 [21]. This approach uses the Markov

chain Monte Carlo approach to estimate age-specific mortality distributions in a Bayesian framework. The 'multibasta' function was used in order to fit the most appropriate underlying mortality model (exponential, Gompertz, Weibull or logistic) and shape (simple, makeham or bathtub) to the data. Models were then compared via deviance information criterion and the best fit model/shape selected (electronic supplementary material, tables S1A and 2A). In each case, four parallel simulations were run (150 000 iterations with a burn-in of 15 001 chains and a thinning of 150). This allowed for robust convergence and low serial auto-correlation (less than 5%; see electronic supplementary material, figures S6 and S7). Differences between posterior distributions of parameter values were then compared across treatment groups by comparing Kullback–Leibler discrepancy calibrations (KLDC). Typically, when comparing KLDC values of model parameters, a value greater than 0.85 (with an upper bound of 1.0) indicates substantial variation in posterior distribution between treatments. Broken stick models, which allowed us to visualize age-specific survival after the dietary switch event, were also run using the same parameters as above apart from setting the minimum age for analysis as 20 (the first-day post-switch).

To assess how various measures of fecundity and mating success changed with dietary treatment, we fitted generalized linear mixed models (GLMMs) using the glmmTMB package v. 1.0.2.1 [22]. For both age-specific fecundity and mating success, we fitted GLMMs including both the linear and quadratic forms of week, dietary treatment and all other higher-order interactions. Vial code was added as a random effect in order to account for repeated measures. While mating success was analysed using a binomial distribution, models with five different error distributions (Poisson, type I negative binomial, type II negative binomial, generalized Poisson and Conway–Maxwell–Poisson) and additional zero-inflation parameters were compared for age-specific fecundity if significant zero-inflation was identified within the residuals of a full Poisson model (using the DHARMa package v. 0.3.2) [23]. The model with the best fit was then chosen by comparing Akaike information criterion (AIC; electronic supplementary material, table S3A). Similar fixed and random effects were then used to test for the effect of diet on LRS (without week and the subsequent interactions). Similar error distributions and zero-inflation parameters were compared if significant zero-inflation was identified, and the model with the best fit was chosen again by AIC (electronic supplementary material, tables S4A and S5A). In both cases, the overall significance of treatment and diet were identified using the Anova function from the car package v. 3.0-9 [24]. Individual fitness ($\lambda ind$) was obtained by calculating the dominant eigenvalue of an age-structured Leslie matrix [25] using the lambda function from the popbio package [26]. One week of pre-reproductive development time was added onto the top-row of the Leslie matrix, denoting age-specific fertility. These values were then analysed with a simple GLM with only treatment as a fixed effect and a Gaussian error distribution. For LRS, mating success and $\lambda ind$ models were first analysed using data across the entire lifespan and then post-switch event. The survival and fitness measures were then visualized with either the ggplot2 [27] or dabestR [28] packages.

## 3. Results

### (a) Effects of continuous and switching diets on age-specific survival

Females that were fed with constant DR or $DR_{od}$ diets had lower baseline mortality rates ($b_0$) than those fed with a constant FF diet (figures 2a and 3a; electronic supplementary material, figures S1 and S2, and table S1B). We also found a significant odour effect, in which the baseline mortality rate of the constant $DR_{od}$ flies was significantly higher than for the constant DR females (electronic supplementary material, figures S1 and S2). However, there was no significant difference between the baseline mortality rates of FF and $FF_{od}$ treatment females (electronic supplementary material, figure S2). With respect to the Gompertz rate parameter ($b_1$), there was no significant difference between constant DR or $DR_{od}$ females and constant FF females (figures 2b and 3b; electronic supplementary material, figure S2). However, Gompertz rate parameter of constant $DR_{od}$ flies was lower than the constant DR females (electronic supplementary material, figure S2). Finally, the Gompertz rate parameter of constant $FF_{od}$ flies was higher than constant FF flies (electronic supplementary material, figure S2).

Diet switching had a direct effect on female mortality. Although the baseline mortality rate of DR-to-FF flies was lower than the constant FF flies after the switch (figure 2c; electronic supplementary material, table S2B), there was an increase in the Gompertz rate parameter to the point that DR-to-FF females were higher than the constant FF females (figure 2d; electronic supplementary material, figure S2). Overall, there was no difference between the post-switch survival curves of DR-to-FF and FF females (electronic supplementary material, figure S3A). In the opposite diet switch comparison, FF-to-DR flies had a significantly lower baseline mortality rate than constant FF flies (figure 2c), but there was no significant difference between the Gompertz rate parameter of FF-to-DR and constant FF females (figure 2d; electronic supplementary material, figure S2). The results were similar in the presence of odour treatments. After the diet switch, $DR_{od}$-to-FF flies had a lower baseline mortality rate but higher Gompertz rate parameter in comparison to the continually FF flies (but not constant $FF_{od}$ flies) (figure 3c,d; electronic supplementary material, figure S2). As a result, there was no difference between the survival curves of $DR_{od}$-to-FF and FF flies (electronic supplementary material, figure S3B). FF-to-$DR_{od}$ flies also had a lower baseline mortality rate than constant FF flies, whereas there was no significant difference between the Gompertz rate parameter of FF-to-$DR_{od}$ and constant FF flies (figure 2; electronic supplementary material, figure S2).

### (b) Effects of continuous and switching diets on mating success

We found significant treatment × week and treatment × week$^2$ interactions for mating success (table 1). In order to better understand the interaction between dietary treatment and age (in weeks), we focused on the change in mating success with age in each dietary treatment separately. Mating success declined gradually with age in all treatments except DR-to-FF and $DR_{od}$-to-FF (figure 4a; electronic supplementary material, tables S8 and S9). In contrast with the other treatments, the switch from DR and $DR_{od}$ to an FF diet prompted a corresponding increase in mating success (figure 4a; electronic supplementary material, tables S8 and S9).

### (c) Effects of continuous and switching diets on fitness

We found significant treatment × week and treatment × week$^2$ interactions for reproductive success (table 1). In order to interpret the interaction between dietary treatment and age, we explored how reproductive success changes with age for each separate dietary treatment. Reproductive success started to decrease after the second week in all flies except those

*Proc. R. Soc. B* **288**: 20211787

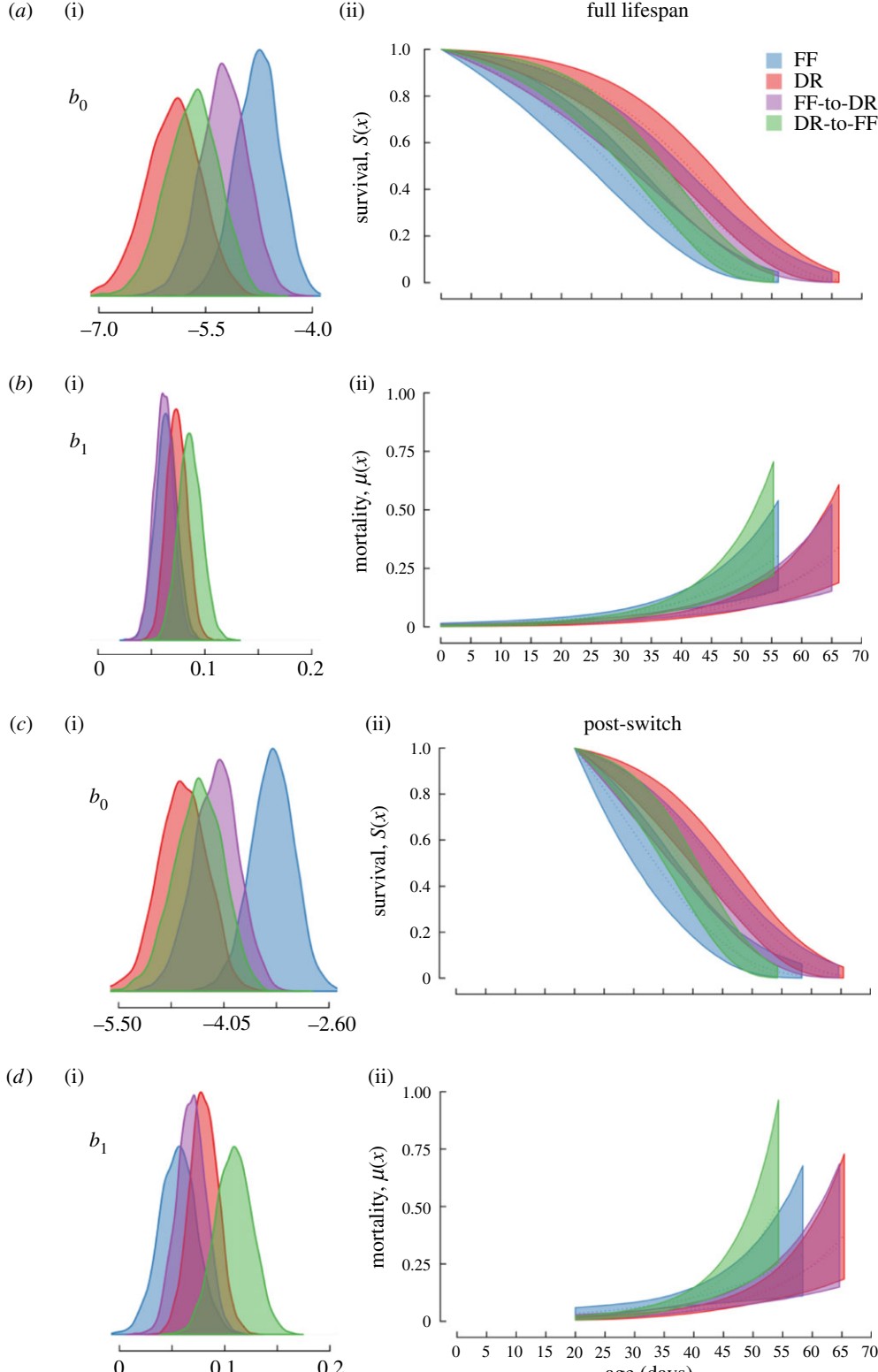

**Figure 2.** Effects of dietary treatments (*without* odour) on age-specific survival and mortality fitted with a simple Gompertz model across the full lifespan (*a*,*b*) or post-switch (*c*,*d*). The posterior distributions of the $b_0$ (baseline mortality rate) and $b_1$ (Gompertz rate parameter) parameters are shown on the (i). Trajectories on the (ii) denote age-specific survival and mortality with the shaded areas representing 95% confidence intervals.

switched from DR/DR$_{od}$ to the FF diet (figure 4*b*; electronic supplementary material, table S3B). In sharp contrast with the other treatments, flies that were switched from DR/DR$_{od}$ to the FF diet showed an immediate response to the switch by laying more eggs. That increase was then followed by an age-related decline similar to the other treatments (figure 4*b*; electronic supplementary material, table S3B).

Dietary treatment also had a significant effect on both LRS and female fitness $\lambda ind$ (table 1). LRS of FF females was similar to those switched from DR-to-FF and FF-to-DR diets, as well as those continually kept on the FF$_{od}$ diet. By contrast, females switched from DR$_{od}$-to-FF or FF-to-DR$_{od}$ diets had slightly lower LRS than the FF diet females. Finally, the LRS of DR and DR$_{od}$ females was significantly lower than for the FF flies (electronic supplementary material, figure S4, and tables S4B, S5B and S6–S7). In terms of individual fitness ($\lambda ind$), there was no significant difference between FF and FF$_{od}$ or FF-to-DR switch females. However, FF-to-DR$_{od}$ switch females had slightly

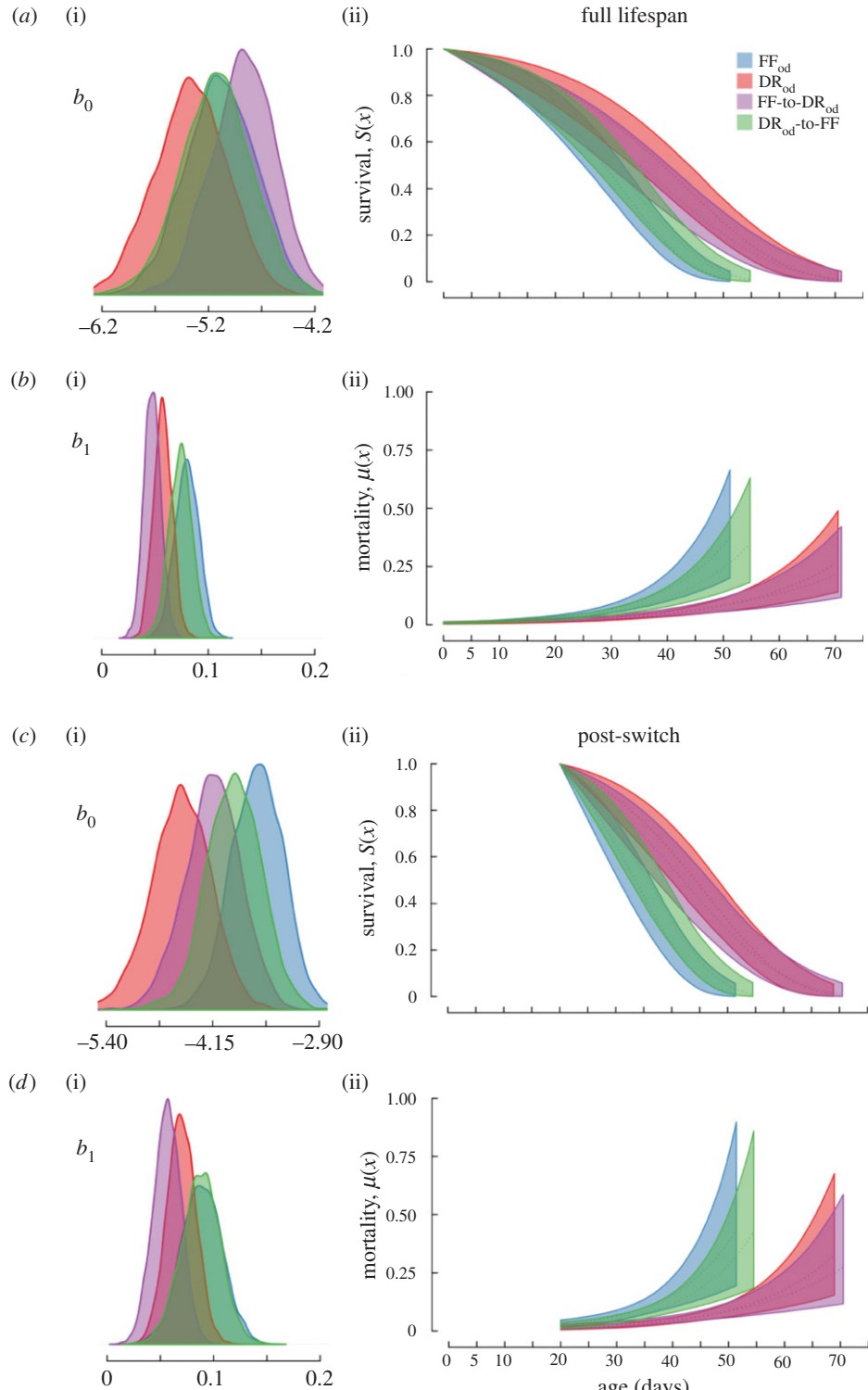

**Figure 3.** Effects of dietary treatment (*with* odour) on age-specific survival and mortality fit with a simple Gompertz model across the full lifespan (*a,b*) or post-switch (*c,d*). The posterior distributions of the $b_0$ (baseline mortality rate) and $b_1$ (Gompertz rate parameter) parameters are shown on the (i). Trajectories on the (ii) denote age-specific survival and mortality with the shaded areas representing 95% confidence intervals.

lower $\lambda ind$ values than FF females. Finally, DR, $DR_{od}$, DR-to-FF switch and $DR_{od}$-to-FF switch females all had lower $\lambda ind$ fitness values than did FF females (electronic supplementary material, figure S5, and tables S4B, S5B and S6–S7).

## 4. Discussion

DR reduced reproduction and increased survival during the days when females were kept on the low-quality diet in comparison to their FF counterparts. However, when DR flies switched back to full feeding (FF), they mated more and showed increased fecundity, though also increased mortality. The post-DR increase in fecundity compensated for decreased reproduction under DR, and the total reproduction of DR-to-FF females was not significantly different from that of the FF flies kept on full feeding throughout their lives. This result shows that the DR response was associated with a marked increase in reproductive output once food access was restored.

Recent studies using inbred lines have suggested that DR might be costly when it is followed by a return to a rich food diet, due to an increase in mortality and decrease in

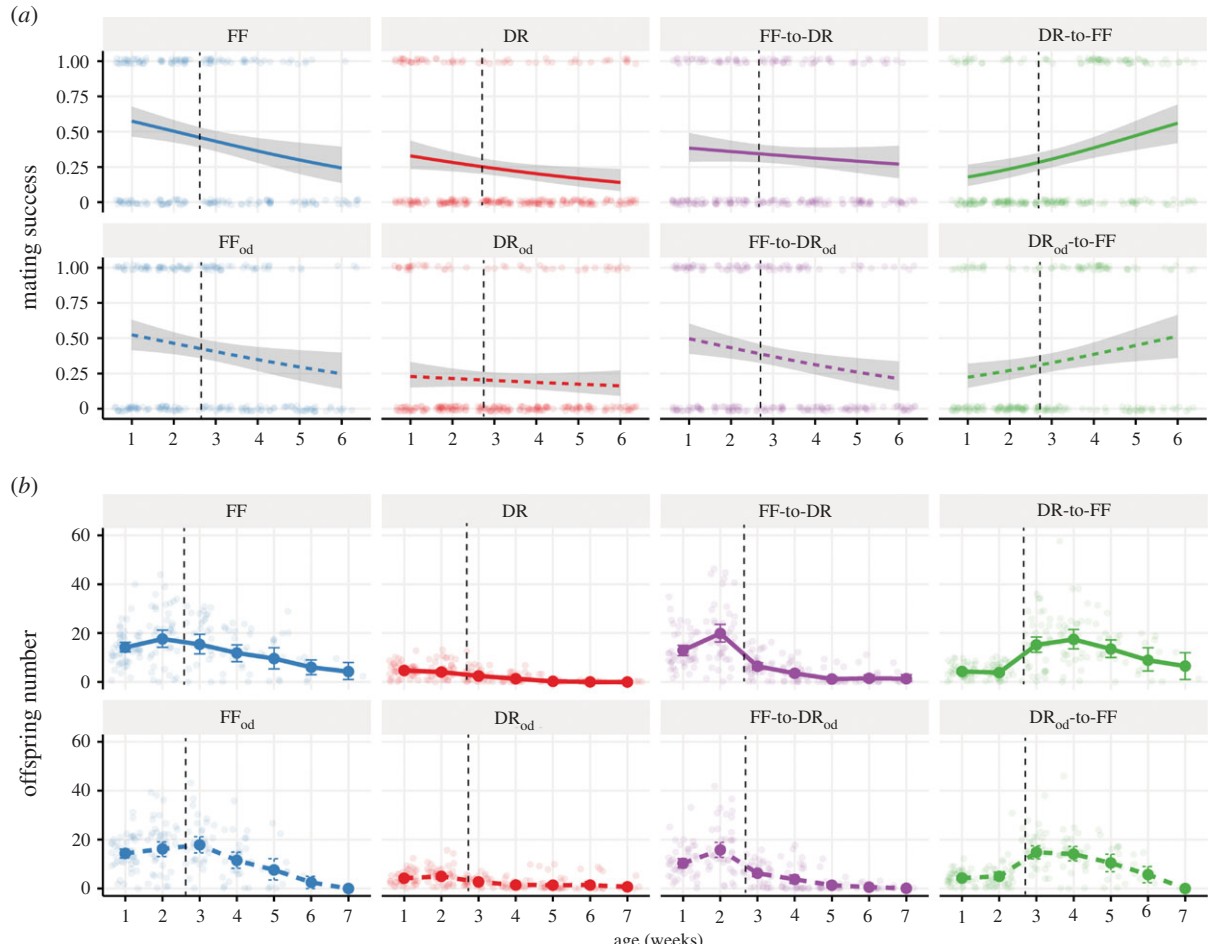

**Figure 4.** Age-related changes in mating success (*a*) and fecundity (*b*) across the different dietary treatments. Individual small points represent raw data while larger points denote average age-specific fecundity with accompanying standard error. For mating success, this is indicated by the shaded grey area around the binomial fit. The vertical dashed line corresponds to the diet-switch time point (day 19).

**Table 1.** Effect of dietary treatment and age (in weeks) on reproductive success, mating success, LRS and lambda ($\lambda$ind).

| response variable | factor | $\chi^2$ | d.f. | *p* |
|---|---|---|---|---|
| age-specific fecundity | treatment × week | 87.085 | 7 | <0.001 |
| | treatment × week$^2$ | 36.414 | 7 | <0.001 |
| mating success | treatment × week | 28.032 | 7 | <0.001 |
| | treatment × week$^2$ | 28.331 | 7 | <0.001 |
| LRS | treatment | 181.85 | 7 | <0.001 |
| $\lambda$ind | treatment | 208.33 | 7 | <0.001 |

reproduction [17]. In this study, we investigated whether similar effects can be observed using a large, outbred population that is well adapted to the environment in which the experimental assays are performed. While our findings support the hypothesis that the DR response is adaptive, there is still much to learn about potential mechanisms underlying the effects of DR on survival, mortality rates and reproduction [17,29]. DR improved survival by reducing the baseline mortality rate ($b_0$) but not the Gompertz rate parameter ($b_1$). This finding is in line with previously published findings of Mair *et al.* [29], which indicated that DR results in an instantaneous improvement in survival and a reduction in mortality rate, while a switch from DR-to-FF immediately increased mortality to the level observed in FF females.

However, in our study, the DR-to-FF switch resulted in a post-switch Gompertz mortality rate that was higher than in control FF flies, a finding that is more in line with the findings of McCracken *et al.* [17]. Because we found a strong increase in mating success and reproduction in the switched females, the most parsimonious explanation is that the increased mortality of DR-to-FF females resulted directly from increased costs of mating and reproduction [30–32].

Even though our findings support a direct trade-off between survival and reproduction, we suggest the mechanism underlying these effects could be more complex. Taken together, the data from Mair *et al.* [29], McCracken *et al.* [17] and our study all suggest that a DR-to-FF switch can result either in a return to a standard FF mortality rate or an

increase in some mortality parameters, depending on the type and the length of DR treatment. The increase in mortality rate can be accompanied by either reduced or increased reproduction, suggesting that the link between increased reproduction and reduced lifespan can be uncoupled. This is further supported by experimental evolution studies suggesting the evolution of reproduction and lifespan under DR is not synchronized, in males [18] or females [7]. Therefore, there is limited support overall, for increased somatic maintenance under DR resulting in reduced damage accumulation and slower ageing, as envisioned by the traditional model [9,10].

Nevertheless, it is possible that DR leads to competition over some other limiting factor rather than energy. For example, recent work in *D. melanogaster* suggests that dietary sterols are one such limiting factor and that DR conditions can result in a trade-off between investment in eggs and soma when dietary cholesterol is insufficient [33]. It is possible that increased egg production on a full diet resulted in the depletion of sterols available for somatic maintenance, and that adding dietary cholesterol in late-life could increase the lifespan of DR-to-FF flies. This is a very interesting possibility and future work is needed to fully investigate the effects of dietary cholesterol on lifespan and fitness. We note, however, that low-protein flies with added cholesterol still outlive high-protein flies with added cholesterol and that cholesterol addition cannot fully compensate for the costs of a high-protein diet [33].

While we focused on DR in female flies in this study, it would be interesting to investigate post-DR survival and reproduction in males. Males and females have been shown to maximize their reproduction on different diets, specifically, different protein–carbohydrate ratios [34], including in *Drosophila* [35]. Therefore, future studies should consider taking sex-specific dietary effects into account. However, we note that males and females can have different dietary preferences [34,36] and, therefore, may have different diets in nature.

Previous studies in *D. melanogaster* and *C. elegans* showed that perception of food availability is sufficient to trigger a reduction of lifespan under DR [13,14]. This suggests that at least part of the DR-driven lifespan extension is under neuronal control and mediated via nutrient-sensing signalling [13,14]. Interestingly, while in *C. elegans* individuals produced fewer eggs under DR than predicted by actual resource availability [16], previous *D. melanogaster* studies suggested that reproduction is largely unaffected by food perception [14]. Despite using a different DR regime to the previous work, we found that food odour increased the baseline mortality of DR flies (but lowered the Gompertz rate parameter), and that continuous exposure to food odour had little effect on overall reproduction and fitness. However, DR$_{od}$-to-FF and FF-to-DR$_{od}$ switch diets had slightly lower reproductive

success than full feeding FF diets, which was not the case with flies on the same switch diets that were unperturbed by exposure to food odour. This suggests that while the presence of food odour altered the perception of the actual dietary environment and increased mortality, the diet switch interacted with food odour in a way that reduced fitness. Importantly, despite the detrimental effect of food odour on mortality rate, the overall response of DR-to-FF flies on odour treatments was qualitatively similar to no-odour treatments. Therefore, they provided additional support to our main findings, suggesting that switching diet from DR-to-FF causes an increase in fitness both in the absence and presence of odour.

Our key finding was that DR increases post-DR mating success and reproduction. The total reproductive success of females that experienced DR during the first two weeks was similar to those kept on full feeding. The switch to full feeding increased the Gompertz rate parameter but not the baseline mortality rate and did not affect overall survival. These results are in line with the hypothesis that the plastic DR response is adaptive. This conclusion is reinforced by the finding that food odour increased mortality rate without affecting reproduction, which suggests that DR flies are in a self-preservation mode under neuronal control. Our understanding of the evolutionary and mechanistic origins of DR response is changing rapidly in the light of recent discoveries. The emerging picture is that DR-driven lifespan extension may result from a combination of several factors, including limiting dietary components [33] and adaptive decision-making in anticipation of better reproductive opportunities in the future. When food becomes available, individuals that have previously been under DR are ready to mate and reproduce to compensate for missed opportunities during the period of famine.

Data accessibility. The data are provided in the electronic supplementary material [37].

Authors' contributions. Z.S.: conceptualization, data curation, formal analysis, funding acquisition, investigation, methodology, visualization, writing-original draft, writing-review and editing; E.R.I.-C.: data curation, formal analysis, investigation, methodology, writing-original draft, writing-review and editing; T.C.: conceptualization, project administration, resources, supervision, writing-original draft, writing-review and editing; A.A.M.: conceptualization, project administration, supervision, writing-original draft, writing-review and editing. All authors gave final approval for publication and agreed to be held accountable for the work performed therein.

Competing interests. The authors declare no conflict of interests.

Funding. This work was funded by BBSRC BB/R017387/1 and ERC GermlineAgeingSoma 724909 to A.A.M. and NERC NE/R010056/1 to T.C.

Acknowledgements. We thank members of the *C. elegans* laboratory and the *Drosophila* laboratory at UEA for fruitful discussions.

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
