## [Peer Review File · Proceedings of the Royal Society B: Biological Sciences]

Review History

RSPB-2021-1787.R0 (Original submission)

Review form: Reviewer 1

Recommendation

Accept with minor revision (please list in comments)

Scientific importance: Is the manuscript an original and important contribution to its field?

Good

General interest: Is the paper of sufficient general interest?

Excellent

Quality of the paper: Is the overall quality of the paper suitable?

Excellent

Is the length of the paper justified?

Yes

Should the paper be seen by a specialist statistical reviewer?

Yes

Do you have any concerns about statistical analyses in this paper? If so, please specify them explicitly in your report.

No

It is a condition of publication that authors make their supporting data, code and materials available - either as supplementary material or hosted in an external repository. Please rate, if applicable, the supporting data on the following criteria.

Is it accessible?

Yes

Is it clear?

Yes

Is it adequate?

N/A

Do you have any ethical concerns with this paper?

No

Comments to the Author

Fitness Benefits of Dietary Restriction

Overall

Sultanova et al have produced a clear and concise manuscript that repudiates the commonly held theory that life extension via dietary restriction (DR) is due to a reallocation of resources from reproduction to somatic maintenance. Instead, the authors suggest that lifespan extension is a result of the reduced direct physiological costs of reproduction and that the impact of DR on lifespan and reproduction can be uncoupled, as suggested through previous works. The authors have designed a straightforward experiment to test whether post-DR, DR worsens reproductive performance comparable to lifetime DR and lifetime fully fed flies. In addition, they tested whether the odour of food, ie neurological signals of "plentiful food" could impact this response. The authors found that, when DR flies were switched to fully fed, reproductive capacity increased to above that of lifetime fully fed flies, resulting in an unchanged reproductive capacity between groups, and this similarly reduced their lifespans back down to that of the fully fed flies. They suggest that this is not due to a "cost" of DR but due directly to the increased cost of mating and reproduction. In addition, they found that DR flies with food odour did have a worse baseline mortality than their DR counterparts.

While I have no major criticisms of this work, I would suggest some changes in the text and figures for clarity. In addition, there appears to be some "lost" supplementary tables and figures that are not mentioned in the text. While the statistics looks sound to me, I am not familiar with all of the algorithms used here so stats consult or review from someone who commonly works with these equations might be helpful.

Finally, I believe there is a need for the addition of diet design and sexual dimorphism in the discussion/introduction. In particular to include some more information on the differences in carbohydrate/protein ratios that have been implicated in both longevity and reproductive success in *Drosophila*. These ratios also impact the reproductive capacity of flies in a sexually dimorphic manner, in particular while males have improved lifespan and reproduction on high carbohydrate low protein, in females lifespan is improved by high carbohydrate but reproduction by protein. This seems pertinent to their proposed theory that lifespan and reproduction are uncoupled, as this may only be true in females.

Minor Comments

Lines 47-51: This section is well written, but not all of the ideas mentioned apply universally to all organisms. It would be helpful to mention the species the research is referring to for each of the studies referenced.

Line 96: Could you explain focal female in the text, I don't think this is a common term across all organisms so it would improve broad understanding if this was defined.

Line 226: There's no FF-DR in Figure S3 but this is referenced in the text.

Line 338: "Despite using a different DR regime to the previous work, we also found that food odour increased the mortality of DR flies" – true but I think figures S1 and S2 could do something to make this clearer, maybe overlaying DR and DRod. This is a super interesting finding and it

would be great if it was clearer in the figure.

Figure 2: For clarity I think this figure would benefit from additional labelling. In particular these figures should be labelled A,B,C and D and references in the text as such which will make it easier to find what is being referenced. Also headings for “full lifespan” and “post-switch” would be helpful. Otherwise you have to read the figure legend every time you glance at the figure.

Figure 3: As with figure 2 I think some additional labelling would and make this clearer and easier to refer to in the manuscript.

Figure S1: It would be helpful to have FF and DR-FF and DR and FF-DR (or vice versa) on the same graphs. Please also add the x-axis labels of days to the top bar of graphs - it's a little difficult to compare them.

Figure S2. The difference between b0 and b1 not explained in figure legend and it also would be nice if they had a non-mathematical label. It would also be easier to reference in the MS if these graphs were labelled A and B.

It is difficult to differentiate between the pairwise comparisons and diet switches as they are represented by – and → could you find a clearer way to represent this.

Figure S3: Is there an explanation for the large drop in survival for DR(od) to FF and particularly FF(od)? It seems quite strange.

I think it would improve clarity if you match colours/line types between Figure S1 and Figure S3. It makes comparing them a little confusing.

Could you explain if the P-value on panel B is the log rank test for trend between all three groups? Or just between FF(od) and DR(od)-FF.

Figure S4: Just wanted to say I like this a lot - especially the unpaired differences - it illustrates your point well.

Figure S6-S8: You have a figure S6-S8 but no reference to it in the text. Additionally, S6 needs a label for the key. Also I am a little confused how Age (Weeks) and Lifespan (Weeks) are different, could you explain this in the legend.

Supplementary Tables: There are lots of supplementary tables not referenced to in the text.

General discussion: It would benefit your discussion and your hypothesis to add some discussion of the potential sexual dimorphism that could be at play here (not explored in your paper but possibly worth repeating in males in the future – maybe a limitation of this study). Jensen et al. found that male and female drosophila have different “optimum” dietary intakes of carbohydrate and protein for reproductive success (and lifespan). This may support your hypothesis that reproduction and lifespan are uncoupled. It is possible that this is only true for females, because in males both lifespan AND reproductive success are optimised by high carbohydrate:low protein but in females while high carbohydrate maximises their lifespan, high protein maximises their reproduction.

This feels similar to some of the results you see here with DR=high carbohydrate and FF=high protein. As you are reducing yeast for your DR, you are mainly constraining protein, which has been shown to be most detrimental to female reproduction but increases lifespan, whereas male reproduction at higher carbohydrate level. When you replace the protein, reproduction increases and lifespan decreases, it would be fascinating to see what happens to males on the same paradigm due to their difference in optima.

<https://www.ncbi.nlm.nih.gov/pmc/articles/PMC4531074/>

Review form: Reviewer 2

Recommendation

Accept with minor revision (please list in comments)

Scientific importance: Is the manuscript an original and important contribution to its field?

Excellent

General interest: Is the paper of sufficient general interest?

Excellent

Quality of the paper: Is the overall quality of the paper suitable?

Excellent

Is the length of the paper justified?

Yes

Should the paper be seen by a specialist statistical reviewer?

No

Do you have any concerns about statistical analyses in this paper? If so, please specify them explicitly in your report.

No

It is a condition of publication that authors make their supporting data, code and materials available - either as supplementary material or hosted in an external repository. Please rate, if applicable, the supporting data on the following criteria.

Is it accessible?

Yes

Is it clear?

Yes

Is it adequate?

Yes

Do you have any ethical concerns with this paper?

No

Comments to the Author

This manuscript by Sultanova and coauthors addresses a fundamental question in life history theory: is the extension of lifespan under dietary restriction adaptive? In this study, they provide evidence that females subjected to dietary restriction fully recover their lifetime reproductive success upon re-exposure to food. The experimental design in this study is excellent, and is supported by sophisticated analyses. Further, the discussion was thorough and very interesting. I thoroughly enjoyed this paper, and have only a few minor comments that I hope might help.

Lines 196-206: I found this text a bit hard to follow. I think this is because it compared the DR and FF treatments in addition to the same diet treatments with odour. The primary figures cited in this paragraph (Figures 2 and 3) don't compare across the treatments with and without odour in this way (this is instead in the supplementary figures). It might make it easier if you discussed the results of figure 2 and 3 (including the diet switching results) first before comparing the no odour/odour treatments from the supp figures.

In the section following Line 241, it would be helpful to remind the reader that this is female mating success. Maybe it's just me, but I always associate mating success with males.

Is the acronym for lifetime reproductive success (LRS) necessary? The paragraph starting on line 269 already has a lot of acronyms, which I find hard to keep track of.

Decision letter (RSPB-2021-1787.R0)

29-Sep-2021

Dear Dr Maklakov:

Your manuscript has now been peer reviewed and the reviews have been assessed by an Associate Editor. The reviewers' comments (not including confidential comments to the Editor) and the comments from the Associate Editor are included at the end of this email for your reference. As you will see, the reviewers and the Editors have raised some concerns with your manuscript and we would like to invite you to revise your manuscript to address them.

Research ethics:

Use of animals and field studies:

It is a condition of publication that you make available the data and research materials supporting the results in the article. Please see our Data Sharing Policies (<https://royalsociety.org/journals/authors/author-guidelines/#data>). Datasets should be deposited in an appropriate publicly available repository and details of the associated accession number, link or DOI to the datasets must be included in the Data Accessibility section of the

article (<https://royalsociety.org/journals/ethics-policies/data-sharing-mining/>). Reference(s) to datasets should also be included in the reference list of the article with DOIs (where available).

If you wish to submit your data to Dryad (<http://datadryad.org/>) and have not already done so you can submit your data via this link [http://datadryad.org/submit?journalID=RSPB&manu=\(Document not available\)](http://datadryad.org/submit?journalID=RSPB&manu=(Document%20not%20available)), which will take you to your unique entry in the Dryad repository.

Please submit a copy of your revised paper within three weeks. If we do not hear from you within this time your manuscript will be rejected. If you are unable to meet this deadline please let us know as soon as possible, as we may be able to grant a short extension.

Best wishes,
Professor Gary Carvalho
mailto: proceedingsb@royalsociety.org

Associate Editor

Comments to Author:

This manuscript has been assessed by two expert reviewers who are both very positive about the novelty and interest of this manuscript, which uses an experimental approach in outbred *Drosophila* to test the adaptive nature of dietary restriction effects on lifespan. I agree with the reviewers on their overall appraisal and they both make a number of suggestions that will strengthen this manuscript further. Reviewer 1 provides some excellent suggestions on how to improve the presentation of results both in the main text and supplement; and also recommends that the introduction and/or discussion could include more background on carbohydrate/protein diet ratio studies in *Drosophila* and sex differences in effects. I appreciate that Proceedings B has strict space constraints but this would be a good contribution - in particular acknowledging that this study has focused on females, and patterns may be different in males. Reviewer 2 makes very minor points which will improve clarity of the manuscript. While also not a fan of unnecessary abbreviations, I appreciate that given the frequency of referring to lifetime reproductive success, in this case it would be acceptable to use the acronym.

On my own reading of the manuscript, I have the following minor suggestions in addition to the reviewer feedback:

l.24 "DR reduces survival and reproduction when nutrients subsequently become plentiful" - I know what you mean here but the phrasing is confusing. Maybe clarify that DR is stage-specific ("DR at an earlier life stage"?)

l.66 Would be helpful to clarify here what are the potential costs that might explain delayed effects of DR on mortality/reproduction when returning to full diet.

l.71 Typo - delete "the" in 'for the developing'

l.85 Would mention in methods section too that flies are outbred (as seems important to mention in the Abstract, Introduction but then non-Drosophila person may wonder what this means in terms of laboratory stock)

l.111 'Excess yeast particles' to return the DR to FF diet seems somewhat vague for study replication: are these measured out to be the same across treatments? (Or is this a standard approach to generating such diets?)

Fig 4 - Use same colour scheme for experimental treatments, and it might be helpful to have vertical line for the point of diet switch on the right-hand graphs.

The authors have provided all data used in the analysis, as required by the journal: it would be helpful to also include a README file explaining meta-data (e.g. the different tables and column descriptions). Importantly, following Proceedings B guidelines, the authors should also make the code available at the point of submission: see <https://royalsociety.org/journals/authors/author-guidelines/#data> Given that the paper describes fairly complex statistics (Bayesian survival analysis), providing the analysis code would be particularly helpful.

Reviewer(s)' Comments to Author:

Referee: 1

Comments to the Author(s)

Fitness Benefits of Dietary Restriction

Overall

Sultanova et al have produced a clear and concise manuscript that repudiates the commonly held theory that life extension via dietary restriction (DR) is due to a reallocation of resources from reproduction to somatic maintenance. Instead, the authors suggest that lifespan extension is a result of the reduced direct physiological costs of reproduction and that the impact of DR on lifespan and reproduction can be uncoupled, as suggested through previous works. The authors have designed a straightforward experiment to test whether post-DR, DR worsens reproductive performance comparable to lifetime DR and lifetime fully fed flies. In addition, they tested whether the odour of food, ie neurological signals of "plentiful food" could impact this response. The authors found that, when DR flies were switched to fully fed, reproductive capacity increased to above that of lifetime fully fed flies, resulting in an unchanged reproductive capacity between groups, and this similarly reduced their lifespans back down to that of the fully fed flies. They suggest that this is not due to a "cost" of DR but due directly to the increased cost of mating and reproduction. In addition, they found that DR flies with food odour did have a worse baseline mortality than their DR counterparts.

While I have no major criticisms of this work, I would suggest some changes in the text and figures for clarity. In addition, there appears to be some "lost" supplementary tables and figures that are not mentioned in the text. While the statistics looks sound to me, I am not familiar with all of the algorithms used here so stats consult or review from someone who commonly works with these equations might be helpful.

Finally, I believe there is a need for the addition of diet design and sexual dimorphism in the discussion/introduction. In particular to include some more information on the differences in carbohydrate/protein ratios that have been implicated in both longevity and reproductive success in Drosophila. These ratios also impact the reproductive capacity of flies in a sexually dimorphic manner, in particular while males have improved lifespan and reproduction on high carbohydrate low protein, in females lifespan is improved by high carbohydrate but reproduction

by protein. This seems pertinent to their proposed theory that lifespan and reproduction are uncoupled, as this may only be true in females.

Minor Comments

Lines 47-51: This section is well written, but not all of the ideas mentioned apply universally to all organisms. It would be helpful to mention the species the research is referring to for each of the studies referenced.

Line 96: Could you explain focal female in the text, I don't think this is a common term across all organisms so it would improve broad understanding if this was defined.

Line 226: There's no FF-DR in Figure S3 but this is referenced in the text.

Line 338: "Despite using a different DR regime to the previous work, we also found that food odour increased the mortality of DR flies" – true but I think figures S1 and S2 could do something to make this clearer, maybe overlaying DR and DRod. This is a super interesting finding and it would be great if it was clearer in the figure.

Figure 2: For clarity I think this figure would benefit from additional labelling. In particular these figures should be labelled A,B,C and D and references in the text as such which will make it easier to find what is being referenced. Also headings for "full lifespan" and "post-switch" would be helpful. Otherwise you have to read the figure legend every time you glance at the figure.

Figure 3: As with figure 2 I think some additional labelling would and make this clearer and easier to refer to in the manuscript.

Figure S1: It would be helpful to have FF and DR-FF and DR and FF-DR (or vice versa) on the same graphs. Please also add the x-axis labels of days to the top bar of graphs - it's a little difficult to compare them.

Figure S2. The difference between b0 and b1 not explained in figure legend and it also would be nice if they had a non-mathematical label. It would also be easier to reference in the MS if these graphs were labelled A and B.

It is difficult to differentiate between the pairwise comparisons and diet switches as they are represented by - and -> could you find a clearer way to represent this.

Figure S3: Is there an explanation for the large drop in survival for DR(od) to FF and particularly FF(od)? It seems quite strange.

I think it would improve clarity if you match colours/line types between Figure S1 and Figure S3. It makes comparing them a little confusing.

Could you explain if the P-value on panel B is the log rank test for trend between all three groups? Or just between FF(od) and DR(od)-FF.

Figure S4: Just wanted to say I like this a lot - especially the unpaired differences - it illustrates your point well.

Figure S6-S8: You have a figure S6-S8 but no reference to it in the text. Additionally, S6 needs a label for the key. Also I am a little confused how Age (Weeks) and Lifespan (Weeks) are different, could you explain this in the legend.

Supplementary Tables: There are lots of supplementary tables not referenced to in the text.

General discussion: It would benefit your discussion and your hypothesis to add some discussion of the potential sexual dimorphism that could be at play here (not explored in your paper but possibly worth repeating in males in the future – maybe a limitation of this study). Jensen et al. found that male and female drosophila have different "optimum" dietary intakes of carbohydrate and protein for reproductive success (and lifespan). This may support your hypothesis that reproduction and lifespan are uncoupled. It is possible that this is only true for females, because in males both lifespan AND reproductive success are optimised by high carbohydrate:low protein but in females while high carbohydrate maximises their lifespan, high protein maximises their reproduction.

This feels similar to some of the results you see here with DR=high carbohydrate and FF=high protein. As you are reducing yeast for your DR, you are mainly constraining protein, which has been shown to be most detrimental to female reproduction but increases lifespan, whereas male reproduction at higher carbohydrate level. When you replace the protein, reproduction increases and lifespan decreases, it would be fascinating to see what happens to males on the same paradigm due to their difference in optima.

<https://www.ncbi.nlm.nih.gov/pmc/articles/PMC4531074/>

Referee: 2

Comments to the Author(s)

This manuscript by Sultanova and coauthors addresses a fundamental question in life history theory: is the extension of lifespan under dietary restriction adaptive? In this study, they provide evidence that females subjected to dietary restriction fully recover their lifetime reproductive success upon re-exposure to food. The experimental design in this study is excellent, and is supported by sophisticated analyses. Further, the discussion was thorough and very interesting. I thoroughly enjoyed this paper, and have only a few minor comments that I hope might help.

Lines 196-206: I found this text a bit hard to follow. I think this is because it compared the DR and FF treatments in addition to the same diet treatments with odour. The primary figures cited in this paragraph (Figures 2 and 3) don't compare across the treatments with and without odour in this way (this is instead in the supplementary figures). It might make it easier if you discussed the results of figure 2 and 3 (including the diet switching results) first before comparing the no odour/odour treatments from the supp figures.

In the section following Line 241, it would be helpful to remind the reader that this is female mating success. Maybe it's just me, but I always associate mating success with males.

Is the acronym for lifetime reproductive success (LRS) necessary? The paragraph starting on line 269 already has a lot of acronyms, which I find hard to keep track of.

Author's Response to Decision Letter for (RSPB-2021-1787.R0)

See Appendix A.

Decision letter (RSPB-2021-1787.R1)

02-Nov-2021

Dear Dr Maklakov

I am pleased to inform you that your manuscript entitled "Fitness Benefits of Dietary Restriction" has been accepted for publication in Proceedings B.

Data Accessibility section

Open Access

Your article has been estimated as being 8 pages long. Our Production Office will be able to confirm the exact length at proof stage.

Paper charges

Sincerely,

Professor Gary Carvalho

Associate Editor:

Board Member

Comments to Author:

(There are no comments.)

Appendix A

Response to Reviewers:

Comments to Author:

This manuscript has been assessed by two expert reviewers who are both very positive about the novelty and interest of this manuscript, which uses an experimental approach in outbred *Drosophila* to test the adaptive nature of dietary restriction effects on lifespan. I agree with the reviewers on their overall appraisal and they both make a number of suggestions that will strengthen this manuscript further. Reviewer 1 provides some excellent suggestions on how to improve the presentation of results both in the main text and supplement; and also recommends that the introduction and/or discussion could include more background on carbohydrate/protein diet ratio studies in *Drosophila* and sex differences in effects. I appreciate that Proceedings B has strict space constraints but this would be a good contribution - in particular acknowledging that this study has focused on females, and patterns may be different in males. Reviewer 2 makes very minor points which will improve clarity of the manuscript. While also not a fan of unnecessary abbreviations, I appreciate that given the frequency of referring to lifetime reproductive success, in this case it would be acceptable to use the acronym.

Thank you very much for this positive and constructive evaluation of our paper. Please find below our point-by-point response to Reviewers' comments. We now point out in the Discussion that the response can indeed be different in males. We also mention the macronutrient approach but we also mention that males and females can have different dietary preferences and, therefore, different diets (LL 398-404).

On my own reading of the manuscript, I have the following minor suggestions in addition to the reviewer feedback:

I.24 "DR reduces survival and reproduction when nutrients subsequently become plentiful" - I know what you mean here but the phrasing is confusing. Maybe clarify that DR is stage-specific ("DR at an earlier life stage"?)

Changed to "early-life DR".

I.66 Would be helpful to clarify here what are the potential costs that might explain delayed effects of DR on mortality/reproduction when returning to full diet.

L 67 we suggest, following [17] that detrimental effects of protein on survival could be one specific cost (it has been suggested previously that an excess of protein can be toxic)

I.71 Typo - delete "the" in 'for the developing'

Done.

I.85 Would mention in methods section too that flies are outbred (as seems important to mention in the Abstract, Introduction but then non-*Drosophila* person may wonder what this means in terms of laboratory stock)

Added to the first sentence (L88).

I.111 'Excess yeast particles' to return the DR to FF diet seems somewhat vague for study replication: are these measured out to be the same across treatments? (Or is this a standard approach to generating such diets?)

Added: "We added eight granules of yeast particles per vial, which is more than flies could consume until they were transferred to a new vial." (LL 118-119)

Fig 4 - Use same colour scheme for experimental treatments, and it might be helpful to have vertical line for the point of diet switch on the right-hand graphs.

Done.

The authors have provided all data used in the analysis, as required by the journal: it would be helpful to also include a README file explaining meta-data (e.g. the different tables and column descriptions). Importantly, following Proceedings B guidelines, the authors should also make the code available at the point of submission: see <https://royalsociety.org/journals/authors/author-guidelines/#data> Given that the paper describes fairly complex statistics (Bayesian survival analysis), providing the analysis code would be particularly helpful.

Done.

Reviewer(s)' Comments to Author:

Referee: 1

Comments to the Author(s)

Fitness Benefits of Dietary Restriction

While I have no major criticisms of this work, I would suggest some changes in the text and figures for clarity. In addition, there appears to be some "lost" supplementary tables and figures that are not mentioned in the text. While the statistics looks sound to me, I am not familiar with all of the algorithms used here so stats consult or review from someone who commonly works with these equations might be helpful.

Thank you, we believe Reviewer referred to the trace plots for Basta analyses. We now mention them in the text (e.g. L 175).

Finally, I believe there is a need for the addition of diet design and sexual dimorphism in the discussion/introduction. In particular to include some more information on the differences in carbohydrate/protein ratios that have been implicated in both longevity and reproductive success in *Drosophila*. These ratios also impact the reproductive capacity of flies in a sexually dimorphic manner, in particular while males have improved lifespan and reproduction on high carbohydrate low protein, in females lifespan is improved by high carbohydrate but reproduction by protein. This seems pertinent to their proposed theory that lifespan and reproduction are uncoupled, as this may only be true in females.

We now refer to this in the Discussion. Indeed, the situation could be different for males and because the two sexes can have very different optimal diets in carbohydrate:protein space, this would have to be taken into account. However, we note that because sexes can have different dietary preferences, they may have very different diets in nature. Studies suggest that even if dietary choice is partly constrained by intra-locus conflict (which is debated), there is still sexual dimorphism in diet choice (Maklakov et al 2008, Rapkin et al 2017).

Minor Comments

Lines 47-51: This section is well written, but not all of the ideas mentioned apply universally to all organisms. It would be helpful to mention the species the research is referring to for each of the studies referenced.

Done.

Line 96: Could you explain focal female in the text, I don't think this is a common term across all organisms so it would improve broad understanding if this was defined.

Changed to "the females".

Line 226: There's no FF-DR in Figure S3 but this is referenced in the text.

Done.

Line 338: "Despite using a different DR regime to the previous work, we also found that food odour increased the mortality of DR flies" – true but I think figures S1 and S2 could do something to make this clearer, maybe overlaying DR and DRod. This is a super interesting finding and it would be great if it was clearer in the figure.

We found that DR(od) has higher baseline mortality rate (b_0) but lower Gompertz's rate parameter (b_1) (See Figure S2) than DR. (LL 410-411)

Figure 2: For clarity I think this figure would benefit from additional labelling. In particular these figures should be labelled A,B,C and D and references in the text as such which will make it easier to find what is being referenced. Also headings for "full lifespan" and "post-switch" would be helpful. Otherwise you have to read the figure legend every time you glance at the figure.

Done

Figure 3: As with figure 2 I think some additional labelling would and make this clearer and easier to refer to in the manuscript.

Done

Figure S1: It would be helpful to have FF and DR-FF and DR and FF-DR (or vice versa) on the same graphs. Please also add the x-axis labels of days to the top bar of graphs - it's a little difficult to compare them.

We added supplementary figures: Figure S8A compares DR-to-FF and FF, Figure S8B compares DRod-to-FF vs. FF; Figure S9A compares FF-to-DR and FF, Figure S9B compares FF-to-DRod and FF.

The X axis label is added in the new SM. See Figure S1.

Figure S2. The difference between b0 and b1 not explained in figure legend and it also would be nice if they had a non-mathematical label. It would also be easier to reference in the MS if these graphs were labelled A and B.

Done.

It is difficult to differentiate between the pairwise comparisons and diet switches as they are represented by – and → could you find a clearer way to represent this.

Done.

Figure S3: Is there an explanation for the large drop in survival for DR(od) to FF and particularly FF(od)? It seems quite strange.

We acknowledge that there was a large drop in survival on these days. The exact reason is not clear.

I think it would improve clarity if you match colours/line types between Figure S1 and Figure S3. It makes comparing them a little confusing.

Done.

Could you explain if the P-value on panel B is the log rank test for trend between all three groups? Or just between FF(od) and DR(od)-FF.

It was comparison of all three groups. If there is no overall difference between the three, there is no need to do individual comparisons.

Figure S4: Just wanted to say I like this a lot - especially the unpaired differences - it illustrates your point well.

Thank you!

Figure S6-S8: You have a figure S6-S8 but no reference to it in the text. Additionally, S6 needs a label for the key. Also I am a little confused how Age (Weeks) and Lifespan (Weeks) are different, could you explain this in the legend.

We removed S6 but left S7 and S8, which are now S6 and S7, as they provide some additional information regarding BASTA analyses.

Supplementary Tables: There are lots of supplementary tables not referenced to in the text.

Done and checked throughout.

General discussion: It would benefit your discussion and your hypothesis to add some discussion of the potential sexual dimorphism that could be at play here (not explored in your paper but possibly worth repeating in males in the future – maybe a limitation of this study). Jensen et al. found that male and female *Drosophila* have different “optimum” dietary intakes of carbohydrate and protein for reproductive success (and lifespan). This may support your hypothesis that reproduction and lifespan are uncoupled. It is possible that this is only true for females, because in males both lifespan AND reproductive success are optimised by high carbohydrate:low protein but in females while high carbohydrate maximises their lifespan, high protein maximises their reproduction.

Thank you for this suggestion, indeed males and females can maximise fitness in different C:P ratios as was shown by Maklakov et al 2008 and later by Jensen et al. 2015 in *Drosophila*. Jensen et al. did not find a sex difference in intake ratio that maximises lifespan but did find a difference in reproduction. We now mention the C:P ratio in the discussion but we also mention that because males and females likely have different dietary preference and different diets, it is not directly relevant to our main conclusion. (LL 396-402)

This feels similar to some of the results you see here with DR=high carbohydrate and FF=high protein. As you are reducing yeast for your DR, you are mainly constraining protein, which has been shown to be most detrimental to female reproduction but increases lifespan, whereas male reproduction at higher carbohydrate level. When you replace the protein, reproduction increases and lifespan decreases, it would be fascinating to see what happens to males on the same paradigm due to their difference in optima.

<https://www.ncbi.nlm.nih.gov/pmc/articles/PMC4531074/>

Yeast has slightly more protein (45%) than carbs (40%) but we cannot say that we mainly constrain protein when we don't add yeast.

Referee: 2

Comments to the Author(s)

This manuscript by Sultanova and coauthors addresses a fundamental question in life history theory: is the extension of lifespan under dietary restriction adaptive? In this study, they provide evidence that females subjected to dietary restriction fully recover their lifetime reproductive success upon re-exposure to food. The experimental design in this study is excellent, and is supported by sophisticated analyses. Further, the discussion was thorough and very interesting. I thoroughly enjoyed this paper, and have only a few minor comments that I hope might help.

Thank you!

Lines 196-206: I found this text a bit hard to follow. I think this is because it compared the DR and FF treatments in addition to the same diet treatments with odour. The primary figures cited in this paragraph (Figures 2 and 3) don't compare across the treatments with and without odour in this way (this is instead in the supplementary figures). It might make it easier if you discussed the results of figure 2 and 3 (including the diet switching results) first before comparing the no odour/odour treatments from the supp figures.

We switched the second and the first paragraphs of the Results.

In the section following Line 241, it would be helpful to remind the reader that this is female mating success. Maybe it's just me, but I always associate mating success with males.

Thank you, we added "female" in the subtitle.

Is the acronym for lifetime reproductive success (LRS) necessary? The paragraph starting on line 269 already has a lot of acronyms, which I find hard to keep track of.

We use LRS several times throughout the text, so it reduces word count, also LRS is an established term in evolutionary biology/ecology so we have retained it to streamline the text. We hope this is OK.